# Homologous Recombination Deficiency (HRD) and BRCA 1/2 Gene Mutation for Predicting the Effect of Platinum-Based Neoadjuvant Chemotherapy of Early-Stage Triple-Negative Breast Cancer (TNBC): A Systematic Review and Meta-Analysis

**DOI:** 10.3390/jpm12020323

**Published:** 2022-02-21

**Authors:** Yue Chai, Yujie Chen, Di Zhang, Yuce Wei, Zhijun Li, Qiao Li, Binghe Xu

**Affiliations:** 1Department of Medical Oncology, National Cancer Center/National Clinical Research Center for Cancer/Cancer Hospital, Chinese Academy of Medical Sciences and Peking Union Medical College, Beijing 100021, China; cy972628990@163.com (Y.C.); irisaaron@163.com (D.Z.); weiyc2311@student.pumc.edu.cn (Y.W.); 15222121700@163.com (Z.L.); xubinghe@medmail.com.cn (B.X.); 2Department of Plastic Surgery, Peking University Third Hospital, Beijing 100021, China; 15738825294@163.com

**Keywords:** homologous recombination deficiency, triple-negative breast cancer, BRCA, platinum agents, neoadjuvant chemotherapy

## Abstract

Background: Platinum-based agents may benefit patients with triple-negative breast cancer (TNBC) whose tumors are dysfunctional in DNA repair mechanisms associated with the homologous recombination repair (HRR) genes. The purpose of this meta-analysis was to assess the values of BRCA1/2 and homologous recombination deficiency (HRD) in the prediction of the pathological complete response (pCR) rates of patients with TNBC treated with platinum-based neoadjuvant chemotherapy (NAC). Patients and Methods: Patients with TNBC with BRCA or HRD status from platinum-based NAC trials were analyzed. The odds ratios (ORs) with 95% confidence intervals (CI) for the identified studies were calculated. Results: 13 eligible studies between January 2000 and September 2021 were included through systematic literature searches of Embase, PubMed, Cochrane, and Web of Science databases. In 12 trials with BRCA status, 629 of 1266 (49.7%) patients with TNBC achieved pCR with platinum-based NAC, including 134 out of 222 (60.4%) BRCA1/2-mutated patients and 495 out of 1044 (47.4%) BRCA wildtype patients (OR, 1.62; 95% CI, 1.20–2.20). The prevalence of HRD was higher than BRCA1/2 mutations in patients with TNBC (69.2% vs. 17.5%). In six trials with HRD information, pCR rates of HRD-positive patients with TNBC were significantly higher than those of HRD-negative patients with TNBC (241/412, 58.5% vs. 60/183, 32.8%, OR, 3.01; 95% CI, 2.07–4.39, *p* < 0.001). Conclusions: BRCA1/2-mutated and HRD-positive patients with TNBC could benefit from platinum-based NAC. In the future, a prospective study using unified HRD testing criteria is warranted for further investigation.

## 1. Introduction

Triple-negative breast cancer (TNBC) refers to a molecular subtype of breast cancer (BC) with negative estrogen receptor, progesterone hormone receptor, and negative human epidermal growth factor receptor 2, accounting for 15–20% of breast tumors [1]. Patients with TNBC had worse prognoses with relatively inferior disease-free survival (DFS) rates after treatment than other BC subtypes, due to their highly invasive and heterogeneous nature. TNBC treatment is still primarily chemotherapy on account of the scarcity of specific molecular targets. Previous studies have shown that tumor-free survival, as well as overall survival of patients with TNBC who underwent neoadjuvant chemotherapy (NAC) and achieved pathological complete response (pCR), is significantly improved, compared with patients without pCR [2,3]. The benefit of platinum-based regimens in patients with early-stage TNBC is controversial because no significant benefits in DFS and overall survival (OS) were observed [4,5]. For this reason, the proper method of selecting a potential marker and further identifying patients with TNBC that can benefit from platinum-based NAC has become one of the research spotlights in recent years.

Homologous recombination deficiency (HRD) prevents the repair of gene damage caused by DNA-damaging agents such as platinum-based chemotherapy, disrupts the ability to homologous recombination (HR), and further causes gene instability [6]. Except for platinum-based drugs, PARP inhibitors and heat shock protein 90(HSP90) inhibitors, for example, are also involved in the homologous recombination repair (HRR) pathway [7,8]. However, the platinum-based regimen remains the most widely used HRD-related regimen for patients with TNBC, due to higher availability and affordability. Germline (g)/somatic(s) BRCA1/2 are the most common cause of HRD. BRCA protein is a type of DNA double-strand damage repair protein encoded by the BRCA gene. BRCA1/2 is a susceptibility gene for both breast and ovarian cancer. Patients with BRCA1/2 mutations comprised about 5.0% of all patients with BC and 3.9% of Chinese patients with BC, respectively [9,10]. The rate of BRCA1/2 mutations was 15–20% in patients with TNBC, which was higher than in other subtypes of BC [11,12,13]. A previous study showed that 69% of patients with BC with BRCA1 mutations fell in the TNBC category, while 16% of patients with BC with BRCA2 mutations fell in the TNBC category [14]. Studies showed that platinum-based regimens in patients with TNBC with BRCA mutations exhibited superior clinical outcomes than those with BRCA wildtype, and scientists deduced that patients with BRCA 1/2 mutations may be sensitive to platinum-based chemotherapy [15,16].

However, over 20% of patients with TNBC harbor HRD without BRCA 1/2 mutations [10,17]. With the exception of BRCA 1/2, other HR-related genes, including PALB2, BARD1, RAD50, RAD51, ATM, MRN, MRE11, RPA, NBS1, CHEK2, TP53, PTEN, and BRIP1, would also increase BC risk and lead to DNA repair deficiencies when these HR genes are mutated or inactivated [18,19]. Previous studies demonstrated that HRD, in addition to being a predictor of PFS and OS of patients who would benefit from platinum-based chemotherapy, is also a major indicator for platinum-sensitive drugs in patients with ovarian cancer [20]. The predictive value of HRD for the efficacy of platinum-based chemotherapy in patients with TNBC with HRD has not been confirmed yet. Therefore, the purpose of this meta-analysis was to assess the values of BRCA 1/2 and HRD in the prediction of the pCR rates of patients with TNBC receiving platinum-based NAC.

## 2. Material and Methods

### 2.1. Literature Search and Selection

Eligible studies were identified through systematic literature searches of Embase, PubMed, Cochrane databases, and Web of Science databases using the date limits January 2000 and September 2021, with no language restriction. Meanwhile, relevant references and guidelines were hand searched, and tracked references also included those of studies with supplementary data. The search strategy was conducted using the following keywords: “breast cancer”, “triple-negative”, “TNBC”, “platinum”, “neoadjuvant chemotherapy”, “BRCA”, “HRD”, “homologous recombination deficiency”, etc. Boolean operators were used for searches, which combined specific keywords and free text terms.

The inclusion criteria of eligible studies were as follows: (1) phase 2 and phase 3 clinical trials, and retrospective studies and cohort studies that included comparisons of TNBC HRD/BRCA 1/2 mutation vs. TNBC non-HRD/BRCA 1/2 wildtype subgroups; (2) patients with TNBC who received platinum-based NAC; (3) those in stage Ⅰ–Ⅲ; (4) available pCR outcomes. Exclusion criteria were (1) case reports; (2) incomplete data on treatment efficacy and available BRCA1/BRCA2/HRD status; (3) other BC subtypes; (4) ongoing studies.

This systematic review and meta-analysis was performed according to the guidelines of the Preferred Reporting Items for Systematic Reviews and Meta-Analyses (PRISMA) [21].

The full protocol is published on the PROSPERO website in November 2021 with the PROSPERO registration number CRD42021279654.

### 2.2. Data Extraction and Quality Assessment

Study selection, extraction of data onto standard forms according to the PRISMA Statement, and cross-checking were conducted independently by two investigators. In case of disagreements, another researcher would be involved in the discussion until resolved. In the process of screening the literature, the title of the literature was read first, and abstracts and full-text were read further to determine eligibility. If necessary, the authors of the original study could be contacted via e-mail or phone for information that had not been confirmed but was relevant to the study. The risk of bias was assessed using the Cochrane Collaboration tool. The quality assessment included selection bias (generating random sequences and blind attribution), attrition bias (selective reporting of incomplete outcome data), performance and selection bias (blinding of participants, healthcare provider, blinding of outcome assessors), and other possible sources of bias in each study. The risk of bias was classified as low, high, or unclear.

Data extracted included the following: (1) basic information of eligible research: research title, first author, published year, study design, trial phase, published journal, ClinicalTrial.gov (accessed on 29 December 2021) number; (2) basic information of the participants: age, number of patients by BRCA mutational status and HRD status, cancer type and stage; (3) intervention measures: platinum type, frequency of administration, dosage, route; (4) number of patients with pCR; (5) key elements of bias risk evaluation.

HRD positive was defined as having a tumor with a predefined HRD score  ≥  cut-off value mentioned in the clinical trial and/or a mutation in BRCA1/2, while HRD negative was an HRD score  < cut-off value and/or BRCA 1/2 wildtype.

### 2.3. Statistical Analysis

The primary endpoint for assessing antitumor efficacy was the pCR rate. pCR was confirmed by pathological evaluation. The definitions of pCR in the included studies are shown in Appendix A. The statistics utilized to compare the treatment effect between two subgroups (BRCA 1/2 mutation vs. BRCA 1/2 wildtype or HRD positive vs. HRD negative) was the odds ratio (OR), with a 95% confidence interval (CI). An OR > 1 indicates a higher pCR rate, whereas an OR < 1 indicates a lower pCR rate in the platinum-based NAC group. A chi-squared test was utilized to compare the differences in pCR rates between the study and control groups.

The results of this study were graphically represented through forest plots. A *p*-value < 0.05 was considered statistically significant. The Higgins I^2^ index was computed to confirm the homogeneity of the study results. A random-effects model (DerSimonian and Laird method) was used when I^2^ is above 50%; otherwise, a fixed-effect model (Mantel–Haenszel) method was used. Funnel plots were used to assess publication bias, and based on their results, publication bias in this study was not statistically significant (Figure 1). Meta-analysis was conducted using Review Manager software (RevMan, version 5.3 for Windows; Cochrane Collaboration, Oxford, UK).

## 3. Results

### 3.1. Study Selection and Characteristics

A total of 1233 articles were primarily identified for preselection. After initial evaluation of each record according to the inclusion and exclusion criteria described, 13 studies were eligible for the meta-analysis (GeparPLA [22], TBCRC030 [23], NCT01525966 [24], NeoSTOP [25], BSMO [26], GeparSixto [27,28], BrighTNess [29,30], PROGECT [31], TBCRC008 [32], PrECOG 0105 [33], NCT01372579 [34], and two clinical trials from Sella et al. in 2018 [35] and Silver et al. in 2010 [36], respectively). The search process is described in Figure 2. The characteristics of the included studies are shown in Table 1 and Table 2.

These 13 studies were divided into 2 subgroups to explore the efficacy of platinum-based NAC for TNBC BRCA-mutated tumors (12 studies with 222 available BRCA1/2-mutated patients and 1044 available BRCA wildtype patients), and for TNBC HRD-positive tumors (6 HRD-predefined studies with 412 HRD-positive patients and 183 HRD-negative patients), respectively.

Analyses of the GeparSixto randomized clinical trials were conducted by Hahnen et al., who explored whether BRCA1/2 mutation status affected treatment response in patients with TNBC in 2017, and by Loibl et al., who investigated the HRD status as a predictor of response in 2018 [27,28]. Similarly, analyses of the BrighTNess randomized clinical trials were conducted by Loibl et al., exploring whether BRCA1/2 mutation status affected treatment response in patients with TNBC, and by Telli et al., who investigated the HRD status as a predictor of response in 2018 [26,27].

### 3.2. Association of Available BRCA1/2 Mutation Status with pCR Rates

In total, 12 studies that reported pCR rates in patients with TNBC were included (Figure 3A). The available BRCA 1/2 mutation rate was 17.5% (222/1266). Of 1266 patients studied, 629 (49.7%) patients with TNBC achieved pCR after platinum-based NAC, and the pCR rates were 60.4% (134/222) and 47.4% (495/1044) in available BRCA1/2-mutated and BRCA wildtype patients, respectively (OR, 1.62; 95% CI, 1.20–2.20). A statistically significant 13.0% increase in the pCR rate was observed in patients with BRCA1/2 mutations (*p*  =  0.002). Trials had moderate heterogeneity (I^2^  = 7%) and were evaluated with a fixed-effect model.

### 3.3. Association of HRD Status with pCR Rates in Four HRD-Predefined Studies

A total of 6 studies with 241 patients had compared pCR rates of HRD-positive patients, with TNBC (*n* = 412), compared with 60 of HRD-negative patients (*n* = 183) (Figure 3B). The definition of HRD positive was different in these six included studies due to different detection methods.

NCT01372579 trial illustrated that 12 patients had HRD-positive results, compared with 14 patients who did not. All patients underwent carboplatin + eribulin. The positive HRD was defined as ≥42, which was derived from the quantitative sum of the telomeric allelic imbalance (TAI), large-scale state transition (LST), and loss of heterozygosity (LOH) scores. Compared with noncarriers, these patients had significantly higher pCR rate (75.0% vs. 14.3%, *p* = 0.02) [34].

GeparSixto trial included patients with TNBC treated with carboplatin + paclitaxel + doxorubicin + bevacizumab. The HRD-positive outcome was defined as an HRD score of ≥42 based on genome-wide copy number and LOH profiling on tumor DNA. In total, 74 patients with TNBC had HRD-positive scores, and 27 patients had HRD-negative scores. Patients with HRD-positive results had higher pCR rates when compared with the matched controls (64.9% vs. 40.7%, *p* < 0.001) [27].

TBCRC 008 included 18 patients with TNBC whose data of HRD and pCR were accessible. Patients were treated with carboplatin + nab-paclitaxel + vorinostat. HRD positive represented an HRD score higher than or equal to a prespecified threshold of 42 without detailed information of the detection method. Overall, 12 patients had HRD-positive scores, and 6 patients had HRD-negative scores. There was no statistically significant difference in pCR between the two groups (66.7% vs. 16.7%, *p* = 0.131) [37].

BrighTNess trial also reported the predictive role of HRD testing on platinum response in 329 patients with TNBC [30]. The HRD threshold of ≥42 vs. <42 and ≥33 vs. <33 were both assessed. When a threshold of ≥42 was exploited to define an HRD-positive group, 138 of 225 HRD-positive patients with TNBC achieved pCR, whereas 42 of 104 HRD-negative patients achieved pCR (*p* < 0.01). However, no relationship between HRD-positive and platinum-sensitive results was observed when an HRD threshold of ≥33 vs. <33 (*p* > 0.05) was used.

PrECOG 0105 trial included 50 patients with TNBC with high HRD–LOH scores and 15 patients with low HRD scores. HRD assessment by HRD–LOH in core breast biopsies before treatment was with a cut-off value of HRD positive of HRD–LOH scores ≥ 10. All patients underwent carboplatin + gemcitabine + iniparib. Patients with high HRD–LOH score had response benefit (residual cancer burden (RCB) of 0/1) to platinum-based NAC, compared with controls (66.0% (33/50) vs. 20.0% (3/15), *p* < 0.01) [33].

TBCRC 030 trial included 56 patients with TNBC whose HRD results were available. Patients were randomized to receive preoperative cisplatin or paclitaxel. Genomic instability was measured by the HRD assay using next-generation sequencing of formalin-fixed paraffin-embedded tumor tissue, with scores of >33 determined to be HRD positive. However, no significant association was observed between HRD scores and pCR rates to either cisplatin or paclitaxel (*p* > 0.05) [23].

The analysis using the fixed-effect model revealed that available HRD-positive patients with TNBC had higher pCR rates in comparison with those of HRD-negative patients (241/412, 58.5% vs. 60/183, 32.8%, OR, 3.01; 95% CI, 2.07–4.39, *p* < 0.001).

## 4. Discussion

In this meta-analysis, we evaluated the roles of BRCA 1/2 and HRD in the prediction of pCR rates of patients with TNBC treated with platinum-based NAC and obtained encouraging results—namely, higher efficacy of platinum-based NAC was observed in BRCA 1/2-mutated/HRD-positive TNBC, compared with BRCA 1/2-wildtype and HRD-negative TNBC.

Platinum, as a DNA-damaging agent, mainly includes cisplatin and carboplatin. The efficacy of platinum-based NAC in the treatment of patients with TNBC is still controversial [38]. In 2019, the Italian Association of Medical Oncology Guidelines on BC supported the addition of platinum to anthracycline/taxane-based NAC for TNBC, which, based on five included studies, increased the probability of pCR rates but not long-term outcomes [39]. BRCA 1/2 gene was one of the topical, proposed, predictive biomarkers for platinum-based regimens in patients with TNBC. Recently, a meta-analysis including seven clinical trials compared the efficacy of platinum-based NAC for patients with TNBC with or without BRCA mutations. The results showed that there was no statistical difference in pCR rates between the two groups of patients (OR = 1.459, 95%C1, 0.95–2.34, *p* = 0.082), which are not in line with our results, for the possible reason of a relatively small study sample, with 159 BRCA-mutated cases [40]. In the current study, 134 out of 222 (60.4%) BRCA1/2 mutated patients and 495 out of 1044 (47.4%) BRCA-wildtype patients achieved pCR (OR, 1.62; 95% CI, 1.20–2.20). A statistically significant 13.0% increase in pCR rate was observed in BRCA1/2-mutated patients (*p*  =  0.002), which confirmed that BRCA status could strongly predict the efficacy of platinum-based chemotherapy.

HRD is regarded as the loss of the ability of cells to repair the DNA double-stranded breaks via HR when DNA double-stranded breaks occur. HRD positive has emerged as a potentially useful biomarker for PARP inhibitors (a DNA single-strand repair inhibitor) in patients with ovarian cancer [41]. Therefore, it could be inferred that HRD may also be a biomarker for the potential benefits of other drugs that induce DNA breaks (such as platinum derivatives, alkylating agents, mitomycin C, etc.) [42]. HRD-related biomarkers are mainly divided into three categories: (1) HRR gene mutations: BRCA1/2 and other HRR genes; (2) genomic scar detection: based on array comparative genomic hybridization, single nucleotide polymorphism (SNP) (TAI, LST, LOH), or mutation signal; (3) transcriptional expression markers detection: transcript analysis, protein expression, function analysis, etc. HRD score, the most commonly used HRD assessment method, is most widely applied based on three SNP sites (TAI, LST, LOH) within the genome. Different HRD scores would be obtained through different detection methods.

The prediction of response to platinum-based agents based on HDR in patients with TNBC remains controversial. The TNT phase 3 clinical trial demonstrated that patients with first-line relapse of TNBC and a BRCA 1/2 mutation who underwent carboplatin showed more favorable objective response rates (68% vs. 33%, *p* = 0.03) and PFS (6.8 vs. 4.4 months, *p* = 0.002) than those of patients who underwent docetaxel [43]. However, such a benefit was not observed in patients with TNBC with BRCA1 methylation, BRCA1 mRNA-low tumors, or a high score in a Myriad HRD assay [43]. The correlation of HRD with pCR rates of patients with TNBC who underwent platinum-based NAC was further investigated in six HRD-predefined studies [23,27,30,32,33,34]. Patients with TNBC with HRD-positive results were inclined to respond to platinum-based NAC according to the results of a meta-analysis in the current study (58.5% vs. 32.8%, OR, 3.01; 95% CI, 2.07–4.39, *p* < 0.001). Patients with TNBC with high HRD scores (≥42 in the GeparSixto, NCT01372579, TBCRCR 008, and BrighTNess study) yielded favorable responses to platinum-based NAC (*p* < 0.05). However, HRD score was not observed as a predictor of pathological response in TNBC in the TBCRC 030 and BrighTNess trial with an HRD score threshold of 33. These results showed that the HRD threshold of 33 had less potential than the HRD threshold of 42 in identifying specified patient populations who might be derived relatively small benefit from platinum-based NAC. Since the scoring method of HRD score was not uniform in various studies, exploring a suitable and universal HRD definition method warrants future investigation. Moreover, the odds ratio for HRD (3.01) as a predictor of pCR rates of patients with TNBC with platinum-based regimens was much higher than the OR for BRCA status (1.62), which suggested that HRD status as a predictor of response to platinum-based NAC in patients with TNBC resulted in larger variation between groups, compared with BRCA 1/2 mutations. In addition, the rate of HRD positive was higher than BRCA 1/2 mutations in patients with TNBC (69.2% vs. 17.5% with an additional detection rate of 51.7%. This also means that nearly half of patients with TNBC may become the dominant group benefiting from platinum-based agents. Only one trial (TBCRC 008) reported that 3 out of 10 (30.0%) HR-positive, HER2-negative luminal B subtype, HRD-positive patients and 1 out of 20 (5.0%) HR-positive, HER2-negative luminal B subtype, HRD-negative patients achieved pCR [32]. The predictive value of HRD for platinum-containing NAC in patients with HR-positive, HER2-negative luminal B subtype could be further explored in the prospective study in the future.

The long-term survival data of patients with TNBC who underwent platinum-based NAC were provided in two included studies in this meta-analysis. GeparSixto trial demonstrated 3-year DFS and 3-year OS rates were higher in patients with TNBC with HRD-positive results who underwent platinum-based NAC than those who were HRD negative (85.7% vs. 77.8; 92.9% vs. 79.9%, *p* < 0.05) [27]. NeoSTOP trial demonstrated that pCR status was significantly related to event-free survival (EFS) and OS (*p* < 0.05) of patients with TNBC who underwent platinum-based NAC on multivariate Cox regression. Estimated 3-year EFS and 3-year OS was 100% and 100% in patients with pCR, compared with 81% and 86% in those without pCR (*p* < 0.003) [25]. It could be inferred that patients with TNBC with BRCA1/2 mutations or HRD positive may be able to obtain survival benefits from platinum-based NAC from the above studies.

No adverse events stratified by BRCA status or HRD status were provided from included studies in this meta-analysis. A meta-analysis published in 2021 involving 14 studies and 3518 patients with TNBC who received NAC or adjuvant chemotherapy proposed that the addition of platinum was related to increased thrombocytopenia and all-grade neuropathy, while neutropenia and grade 3–4 neuropathy did not increase [4].

Nevertheless, HRD scores with different candidate HR genes were calculated by different methods that differed from institute to institute in these four studies, which may have a slight influence on the results [23,27,32,34]. Second, different trials with different choices of chemotherapy agents might bias results. The chemotherapeutic agents used in our enrolled trials included PARP inhibitors, taxanes, anthracyclines, alkylating agents, and platinum. However, only platinum and PARP inhibitors were most associated with the HRR pathway. Of the six trials that had compared pCR rates of HRD-positive patients with TNBC with those of HRD-negative patients, BrighTNess and PrECOG 0105 trials included PARP inhibitors and platinum simultaneously, and we still included these two studies in this meta-analysis due to relatively limited data of HRD in the prediction of the pCR rates of patients with TNBC currently treated with platinum-based NAC. Thirdly, the definitions of pCR, treatment modalities, evaluation criteria, and the technologies to access tumors were not unified in the included studies, although with modest differences, which might cause a certain bias. Fourth, the BRCA1/2 mutation is currently the most common mutation known to cause HRD. As there are relatively few studies (only six trials enrolled in this meta-analysis) that explored the predictive value of HRD for the efficacy of platinum-based NAC in patients with TNBC, and no available data were provided on the response to platinum-based regimens in patients with HRD but without BRCA1/2 mutation in these six studies, the conclusion was based on patients with HRD (with/without BRCA1/2 mutation). The conclusions of our studies could be verified in patients with HRD but without BRCA1/2 mutation in the future. Fifth, the difference in adverse reactions to platinum-based agents between patients stratified by HRD could not be analyzed due to the loss of documents in the included studies. A prospective study with unified HRD testing criteria is warranted in the future.

## 5. Conclusions

BRCA1/2-mutated and HRD-positive patients with TNBC could benefit from platinum-based NAC. A prospective study with HRD testing criteria is warranted in the future.

## Figures and Tables

**Figure 1 jpm-12-00323-f001:**
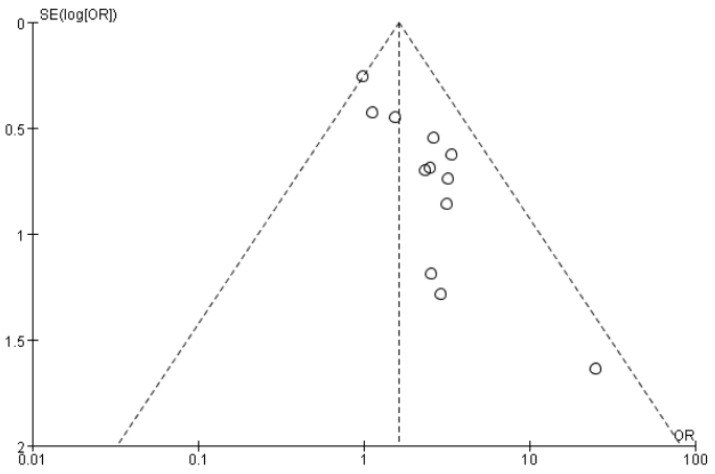
Funnel plot showing publication bias among the studies included in the analysis.

**Figure 2 jpm-12-00323-f002:**
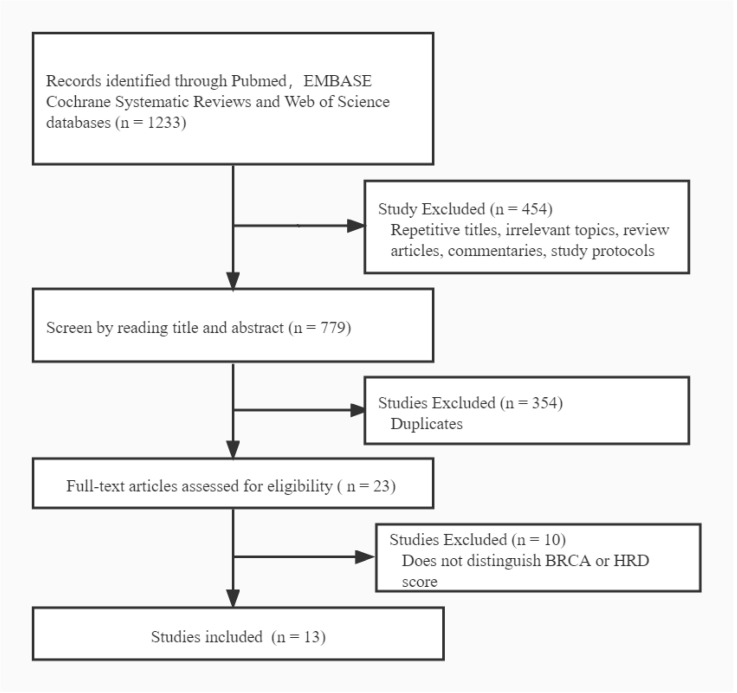
Flowchart of the literature search.

**Figure 3 jpm-12-00323-f003:**
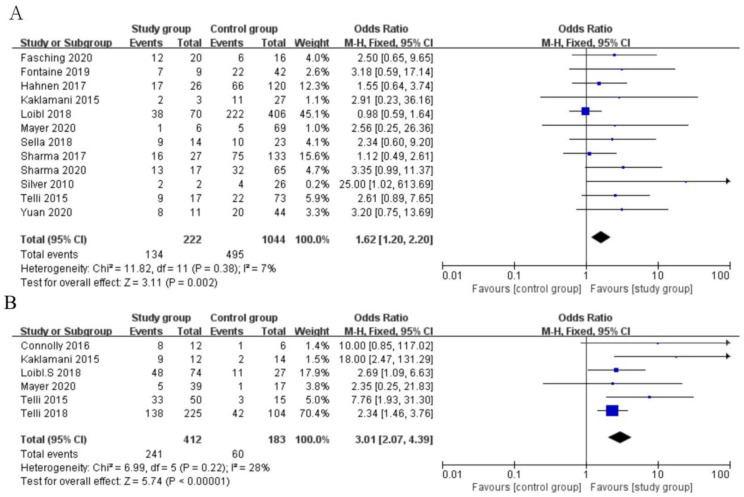
Forest plot for comparison of pCR rates: (**A**) patients with BRCA 1/2 mutation vs. patients with BRCA1/2 wildtype; (**B**) HRD-positive vs. HRD-negative patients in 6 HRD-predefined studies.

**Table 1 jpm-12-00323-t001:** Characteristics of the included studies.

Author	Study	Year	Type of Study	Stage of Disease	Endpoints Available for Inclusion	Treatment	Chemotherapy Regimen
Fasching et al. [22]	GeparPLA	2020	Subgroup of phase 2 RCT	I-III	pCR; toxicity	PCb→EC	P 80 mg/m^2^ weekly + Cb AUC2 weekly for 12 weeks, followed by EC.
Mayer et al. [23]	TBCRC030	2020	Subgroup of phase 2 RCT	I-III	pCR; RCB	Cisplatin	Cisplatin 75 mg/m^2^ every 3 weeks
Yuan et al. [24]	NCT01525966	2020	Phase 2	II-III	pCR; RCB; 3-year OS; 3-year DFS	Cb + nab-P	P 80 mg/m^2^ weekly for 12 doses + Cb AUC 6 every 3 weeks for four cycles
Sharma et al. [25]	NeoSTOP	2020	Phase 2 RCT	I-III	pCR; RCB; OS; toxicity; event-free	Arm A: PCb→AC; Arm B: DCb	Arm A: P 80 mg/m^2^ weekly for 12 weeks + Cb AUC6 every 3 weeks for four cycles followed by doxorubicin 60 mg/m^2^ + cyclophosphamide 600 mg/m^2^ every 14 days for four cycles Arm B: D 75 mg/m^2^ + Cb AUC6 every 3 weeks for six cycles
Fontaine et al. [26]	BSMO	2019	phase 2	II-III	pCR; toxicity	PCb→EC	P 80 mg/m^2^ weekly concurrent with weekly Cb AUC = 2 for 12 weeks, followed by bi-weekly epirubicin (90 mg/m^2^) and cyclophosphamide (600 mg/m^2^)
Hahnen et al. [28], Loibl.S et al. [27]	Gepar Sixto	2017, 2018	Phase 2 RCT	II-III	pCR, DFS	PCb + doxorubicin + bevacizumab	Cb AUC5 + P 80 mg/m^2^ + doxorubicin 20 mg/m^2^ weekly for 18 weeks + bevacizumab 15 mg/kg iv every 3 weeks
Loibl et al. [29], Telli et al. [30]	BrighTNess	2018	phase 3 RCT	II-III	pCR, toxicity	Segment I: PCb + veraparib Segment II: PCb	Segment I: P 80 mg/m^2^ weekly for 12 doses + Cb AUC 6 every 3 weeks for four cycles + veraparib 50 mg orally twice a day. Segment II: P 80 mg/m^2^ weekly for 12 doses + Cb AUC 6 every 3 weeks for four cycles
Sella et al. [35]		2018	Clinical trial	I-III	pCR	ddAC→PCb	Four cycles of doxorubicin (60 mg/m^2^) and cyclophosphamide (600 mg/m^2^) every 2 weeks followed by 12 weekly cycles of P (80/m^2^) with Cb (AUC 1.5)
Sharma et al. [31]	PROGECT	2017	Clinical trial	I-III	pCR, RCB	DCb	Six cycles of Cb AUC 6 + D 75 mg/m^2^ every 21 days
Connolly et al. [32]	TBCRC 008	2016	Phase 2 RCT	I-III	pCR	Cb + nab-P ± vorinostat	Not available
Telli et al. [33]	PrECOG 0105	2015	Phase 2	I-IIIA	pCR, RCB	Cb + gemcitabine + iniparib	Four cycles of Cb (on days 1 and 8) + gemcitabin (1000 mg/m^2^ on days 1 and 8), + iniparib (5.6 mg/Kg on days 1, 4, 8, and 11) every 21 days
Kaklamani et al. [34]	NCT01372579	2015	Phase 2	I-III	pCR, RCB	Cb + eribulin	Four cycles of Cb AUC 6 + eribulin 1.4 mg/m^2^ (day 1 and 8) every 21 days
Silver et al. [36]		2010	Clinical trial	II-III	pCR	Cisplatin	Four cycles of Cisplatin at 75 mg/m^2^ every 21 days

Abbreviations: RCT: randomized controlled trial; pCR: pathological complete response; P: paclitaxel; Cb: carboplatin; EC: epirubicin/cyclophosphamide; RUC: area under the curve; RCB: residual cancer burden; OS: overall survival; DFS: disease-free survival; nab-P: nab-paclitaxel; AC: doxorubicin/cyclophosphamide; D: docetaxel; DCb: docetaxel/carboplatin; dd: dose dense.

**Table 2 jpm-12-00323-t002:** The number of patients with BRCA or HRD status in the included studies.

Author	Study	Year	No. of Patients				No. of Patients with pCR			
			mBRCA+	WtBRCA	HRD+	HRD-	mBRCA+	WtBRCA	HRD+	HRD-
Fasching et al. [22]	GeparPLA	2020	20	16	27		12	6	16	
Mayer et al. [23]	TBCRC030	2020	6	69	39	17	1	5	5	1
Yuan et al. [24]	NCT01525966	2020	11	44			8	20		
Sharma et al. [25]	NeoSTOP	2020	17	65			13	32		
Fontaine et al. [26]	BSMO	2019	9	42			7	22		
Hahnen et al. [28], Loibl.S et al. [27]	GeparSixto	2017, 2018	26	120	74	27	17	66	48	11
Loibl et al. [29], Telli et al. [30]	BrighTNess	2018	70	406	225	104	38	222	138	42
Sella et al. [35]		2018	14	23			9	10		
Sharma et al. [31]	PROGECT	2017	27	133			16	75		
Connolly et al. [32]	TBCRC 008	2016			12	6			8	1
Telli et al. [33]	PrECOG 0105	2015	17	73	50	15	9	22	33	3
Kaklamani et al. [34]	NCT01372579	2015	3	27	12	14	2	11	9	2
Silver et al. [36]		2010	2	26			2	4		

Abbreviations: HRD: homologous recombination deficiency; pCR: pathological complete response CR: complete response; mBRCA: mutated BRCA; WtBRCA: wildtype BRCA.

## Data Availability

The datasets developed and analyzed in this study are available from the corresponding author upon reasonable request.

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
