# Peer review of "Homologous Recombination Deficiency (HRD) and BRCA 1/2 Gene Mutation for Predicting the Effect of Platinum-Based Neoadjuvant Chemotherapy of Early-Stage Triple-Negative Breast Cancer (TNBC): A Systematic Review and Meta-Analysis"

_jpm, 2022, doi:10.3390/jpm12020323_

Round 1

Reviewer 1 Report

Yue Chai et al have done review of literature and meta-analysis to show Homologous recombination deficiency (HRD) and BRCA1/2 gene mutation for predicting the effect of platinum-based neo-adjuvant chemotherapy of early-stage triple-negative breast 4 cancer (TNBC). The hypothesis is clearly stated, and the methodology/analysis is done as per standard guidelines. The study is interesting but there are few concerns which must be addressed.

Comments:

  • BRCA1 deficient tumors are predominantly TNBC whereas BRCA2 are not. BRCA2 tumors have no proper correlation with hormonal status, or it is predominantly ER positive, PR positive, HER2 negative Luminal B subtype tumors. So, just analyzing TNBC with BRCA2 status might result in inaccuracy in the analysis.
  • After screening many trials, finally 6 studies are eligible for the analysis. It will be very difficult to conclude based on just 6 studies. It will be good if HRD positive non-TNBC trials, if available, are considered for analysis which will act as a control and strengthen the current observations.

Author Response

Response: Thank you for your valuable suggestion. Only one trial (TBCRC 008) that we found reported that 3 out of 10 (30.0%) HR positive, HER2 negative luminal B subtype, HRD positive and 1 out of 20 (5.0%) HR positive, HER2 negative luminal B subtype, HRD negative patients achieved pCR. It also showed that the predictive value of HRD for platinum-based NAC in patients with HR positive, HER2 negative luminal B subtype was similar to patients with TNBC. We have added this information in the discussion part (page 11, lines 292-299). There were few studies that provided the data of HRD in the prediction of the pCR rates of patients with non-TNBC treated with platinum-based NAC. We are conducting a study on the predictive value of HRD in advanced breast cancer and look forward to results. Therefore, the predictive value of HRD for platinum-containing NAC in patients with HR positive, HER2 negative luminal B subtype could be further explored in the prospective study in the future.

Reviewer 2 Report

1) how do you define or answer for different trials and their different chemo agent choices to treat BC patient and your outcome?

2) Definition of PCR is also defined differently in different trials, how would you account for that?

3) Yes, your conclusion seems to be in right direction but there are multiple factors since the chemo choices from 2010 to 2020 is different and the technology they have used to access tumor is also different. How sure are you about each approach they have taken in these trials? 

Author Response

Point 1: how do you define or answer for different trials and their different chemo agent choices to treat BC patient and your outcome?

Response 1: We agree with your suggestion. Different trials with different chemotherapy agent choices might bias results. The chemotherapeutic agents used in our enrolled trials included PARP inhibitors, taxanes, anthracyclines, alkylating agents, and platinum. However, only platinum and PARP inhibitors were most associated with the HRR pathway. Of the 6 trials which had compared pCR rates of HRD positive patients with TNBC compared with HRD negative patients, BrighTNess and PrECOG 0105 trials included PARP inhibitors and platinum simultaneously, and we still included these two studies in this meta-analysis due to relatively limited data of HRD in the prediction of the pCR rates of patients with TNBC treated with platinum-based NAC. This is indeed a shortcoming of this article. So this limitation of this study has been added in the Discussion part (page 11-12, lines 318-325).

Point 2: Definition of PCR is also defined differently in different trials, how would you account for that?

Response 2: Thank you for this valuable suggestion. We acknowledge that the definition of pCR varies slightly between the enrolled studies. For ease of understanding, the detailed definition of pCR in each included study could be found in Supplemental Table 1(page 3, lines 129-130). From this table, we could find that pCR was mostly defined as no residual invasive tumor in breast and in axillary lymph nodes (ypT0/is ypN0), and changes in different studies were made based on this definition. This limitation of this study was also added in the Discussion part (page 12, lines 326-328).

Point 3: Yes, your conclusion seems to be in right direction but there are multiple factors since the chemo choices from 2010 to 2020 is different and the technology they have used to access tumor is also different. How sure are you about each approach they have taken in these trials? 

Response 3: We agree with your suggestion. From 2010 to 2020, different studies employed different treatment modalities, evaluation criteria, and detection techniques depending on the corresponding years. However, in terms of chemotherapy regimens, we could see that the dosage and method of administration in each study were according to breast cancer guidelines, and no study has been found that the drug dosage was significantly too high or too low. Similarly, the evaluation criteria and detection techniques of each included literature were also established according to the guidelines of the corresponding year. Of course, we still acknowledge that this is another shortcoming in this study and therefore added it in the discussion part (page 12, lines 326-328). Your opinions made this article get significant improvement. Thank you very much!

Please see the attachment (Cause only one word file can be selected to reply to you on this website,  I uploaded the revised full text. Supplemental Table 1 I will send to you through Editor). Thank you again!

Reviewer 3 Report

This is an interesting study since there is documented benefit of platinum based chemotherapy for patients with BRCA1/2 gemline mutations. It is therefore obvious that TNBC with HRD deficiency without detected BRCA1/2 mutations could also benefit from this treatment.

The findings suggest that there is a benefit for this patient group from Platin. However, a major question in the analysis is if the conclusion is based on patients with HRD but no BRCA1&2 mutation? Apparently some of the HRD studies did not screen for mutations and therefore the results may be inflated by BRCA mutation mediated HR deficiency. In figure 3 B: Do these studies include BRCA1/2 mutated patients?

Minor things:

Table 2. Wrong reference for the Hahnen study, should be 15 not 24.

Line 172, reference for Loibl missing.

Line 271: ORR not definies

Author Response

Point 1: This is an interesting study since there is documented benefit of platinum based chemotherapy for patients with BRCA1/2 gemline mutations. It is therefore obvious that TNBC with HRD deficiency without detected BRCA1/2 mutations could also benefit from this treatment.

The findings suggest that there is a benefit for this patient group from Platin. However, a major question in the analysis is if the conclusion is based on patients with HRD but no BRCA1&2 mutation? Apparently some of the HRD studies did not screen for mutations and therefore the results may be inflated by BRCA mutation mediated HR deficiency. In figure 3 B: Do these studies include BRCA1/2 mutated patients?

Response 1: We agree with your comment. The BRCA 1/2 mutation is the most common mutation known to cause HRD currently. HRD positive was defined as if a tumor’s a predefined HRD score ≥ cut-off value mentioned in the clinical trial and/or a mutation in BRCA1/2 while HRD negative was an HRD score  < cut-off value and/or BRCA 1/2 wildtype. Because there are relatively few studies (only 6 trials enrolled in this meta-analysis) that explored the predictive value of HRD for the efficacy of platinum-based NAC in patients with TNBC, and no available data on the response to platinum-based regimens in patients with HRD but without BRCA1/2 mutation were provided in these 6 studies, the conclusion in this study was based on patients with HRD with/without BRCA1/2 mutation. In figure 3 B: these studies include BRCA1/2 mutated patients. The conclusions of our studies could be verified in patients with HRD but without BRCA1/2 mutation in the future. This limit has been added in the discussion part (page 12, lines 328-334). Eligible studies were identified through systematic literature searches of Embase, Pubmed, Cochrane databases, and Web of Science databases using the date limits January 2000 and September 2021, with no language restriction.

Point 2: Minor things:

1) Table 2. Wrong reference for the Hahnen study, should be 15 not 24.

2) Line 172, reference for Loibl  missing.

3) Line 271: ORR not definies

Response 2: Thank for your valuable suggestion. 1) have changed the reference for the Hahnen study 24 to 25(table 1 and table 2). 2) have added to the study (page 8, lines 172 and 175). 3) have added to the study (page 10, lines 270). Your opinions made this article get significant improvement. Thank you very much!

Round 2

Reviewer 1 Report

Comments:

  • The response to the comment is not satisfactory. It is very difficult to argue that the predictive value of HRD for platinum-based NAC in patients with HR positive, HER2 negative luminal B subtype was similar to patients with TNBC. It could be an inaccurate prediction because of limited sample number. If the number of studies is limited as per authors response, the authors must at least properly explain how much percentage of BRCA2 defective tumors falls in TNBC/HRD category in introduction section. 
  • New treatment regimens available for TNBC in line 61 (for example: 17AAG in clinical trials (sengodansk et al, carcinogenesis 2019; Pramod P. Mehta et al, 2011, clinical cancer research etc.)) must be elaborated and establish the why platinum-based NAC/pCR is important for HRD/TNBC in introduction section.
  • Similarly, HRD must be clearly defined in introduction. All the genes which contribute to HRD category other than BRCA1/2 must be briefly elaborated (for example, MRN, RAD51 etc)

Author Response

Response:

Thank you for your valuable suggestion. We have deleted the sentence “It also showed that the predictive value of HRD for platinum-based NAC in patients with HR positive, HER2 negative luminal B subtype was similar to patients with TNBC.”  because it may be not accurate (page 11, lines 309-313). And because there are relatively few studies (only 6 trials enrolled in this meta-analysis) that explored the predictive value of HRD for the efficacy of platinum-based NAC in patients with TNBC, no available data on the number/percentage of patients with HRD and BRCA2 mutation were provided simultaneously in studies that we could search online currently. But we added that how much percentage of BRCA2 defective tumors falls in TNBC category but not TNBC/HRD in introduction section (A previous study has shown that 69% of patients with BC with BRCA1 mutations fell in TNBC category, while 16% of patients with BC with BRCA2 mutations fell in TNBC category), which may lead the content of this article more clear. (page 2, lines 74-76).

The main aim of our meta-analysis was to explore the predictive value of HRD for the efficacy of platinum-based chemotherapy in patients with TNBC with HRD, so we did not describe the role of HRD in patients with HR positive, HER2 negative luminal B subtype breast cancer.

  To our knowledge, 17AAG in clinical trials published by Pramod P. Mehta et al (2011, clinical cancer research) has been retracted by the journal in 2017 (Retraction: effective targeting of triple-negative breast cancer cells by pf-4942847, a novel oral inhibitor of hsp 90, Clin. Cancer Res. 23 (2017) 612,

https://doi.org/10.1158/1078-0432.CCR-16-2872.). It can be seen that the role of 17AAG in TNBC is still in the exploratory stage. Except for HSP90 inhibitors, PARP inhibitors were also involved in homologous recombination repair (HRR) pathway, and we have added them both in introduction section (page 2, lines 64-66). The relationship between platinum-based neoadjuvant chemotherapy and HRD in TNBC is currently the most explored in RCTs compared to other HRR-related drugs, so we chose platinum-based drugs for this meta-analysis. In addition, platinum-based regimen remains the most widely used HRD-related regimen for patients with TNBC due to its higher availability and affordability. Therefore, platinum-based NAC/pCR is important for HRD/TNBC (page 2, lines 66-68). We also added this information in the introduction of our manuscript.

   According to your suggestion, the main genes which contribute to HRD category other than BRCA1/2 have been briefly added in the introduction section. Because gene panels of HRD in different institutions were different (this shortcoming has been added to the article in the discussion part)and there are many HRD-related genes, so we only listed the main HRD-related genes(page 2, lines 82-83).

    Your valuable opinions made this article get significant improvement. Thank you very much again!

Reviewer 2 Report

Thank you for addressing my comments. I have no further comments.

Author Response

Thank you very much again!

Reviewer 3 Report

the manuscript is acceptable now. 

Author Response

Thank you very much again!